# Demographic and Socioeconomic Factors in Prospective Retina-Focused Clinical Trial Screening and Enrollment

**DOI:** 10.3390/jpm13060880

**Published:** 2023-05-23

**Authors:** Jessica A. Cao, Sagar B. Patel, Calvin W. Wong, David Garcia, Jose Munoz, Cassandra Cone, Deneva Zamora, Mary Reagan, Tieu V. Nguyen, Will Pearce, Richard H. Fish, David M. Brown, Varun Chaudhary, Charles C. Wykoff, Kenneth C. Fan

**Affiliations:** 1Retina Consultants of Texas, Houston, TX 77401, USA; jessica.cao@retinaconsultantstexas.com (J.A.C.); sbpmd@retinaconsultantstexas.com (S.B.P.); jose.munoz@retinaconsultantstexas.com (J.M.); cassandra.cone@retinaconsultantstexas.com (C.C.); deneva.zamora@retinaconsultantstexas.com (D.Z.); mary.reagan@retinaconsultantstexas.com (M.R.); vtnmd@retinaconsultantstexas.com (T.V.N.); rhfmd@retinaconsultantstexas.com (R.H.F.); dmbmd@retinaconsultantstexas.com (D.M.B.); ccwmd@retinaconsultantstexas.com (C.C.W.); 2Blanton Eye Institute, Houston Methodist Hospital & Weill Cornell Medical College, Houston, TX 77030, USA; 3McGovern Medical School, The University of Texas Health Science Center at Houston, Houston, TX 77030, USA; calvin.w.wong@uth.tmc.edu; 4Department of Surgery, McMaster University, Hamilton, ON L8S 4L8, Canada; vchaudh@mcmaster.ca

**Keywords:** demographic, socioeconomic, retina-focused clinical trial

## Abstract

Historically marginalized populations are disproportionately affected by many diseases that commonly affect the retina, yet they have been traditionally underrepresented in prospective clinical trials. This study explores whether this disparity affects the clinical trial enrollment process in the retina field and aims to inform future trial recruitment and enrollment. Age, gender, race, ethnicity, preferred language, insurance status, social security number (SSN) status, and median household income (estimated using street address and zip code) for patients referred to at least one prospective, retina-focused clinical trial at a large, urban, retina-based practice were retrospectively extracted using electronic medical records. Data were collected for the 12-month period from 1 January 2022, through 31 December 2022. Recruitment status was categorized as Enrolled, Declined, Communication (defined as patients who were not contacted, were contacted with no response, were waiting for a follow-up, or were scheduled for screening following a clinical trial referral.), and Did Not Qualify (DNQ). Univariable and multivariable analyses were used to determine significant relationships between the Enrolled and Declined groups. Among the 1477 patients, the mean age was 68.5 years old, 647 (43.9%) were male, 900 (61.7%) were White, 139 (9.5%) were Black, and 275 (18.7%) were Hispanic. The distribution of recruitment status was: 635 (43.0%) Enrolled, 232 (15.7%) Declined, 290 (19.6%) Communication, and 320 (21.7%) DNQ. In comparing socioeconomic factors between the Enrolled and Declined groups, significant odds ratios were observed for age (*p* < 0.02, odds ratio (OR) = 0.98, 95% confidence interval (CI) [0.97, 1.00]), and between patients who preferred English versus Spanish (*p* = 0.004, OR = 0.35, 95% CI [0.17, 0.72]. Significant differences between the Enrolled and Declined groups were also observed for age (*p* < 0.05), ethnicity (*p* = 0.01), preferred language (*p* < 0.05), insurance status (*p* = 0.001), and SSN status (*p* < 0.001). These factors may contribute to patient participation in retina-focused clinical trials. An awareness of these demographic and socioeconomic disparities may be valuable to consider when attempting to make clinical trial enrollment an equitable process for all patients, and strategies may be useful to help address these challenges.

## 1. Introduction

Common retina diseases such as diabetic retinopathy (DR), diabetic macular edema (DME), age-related macular degeneration (AMD), and retinal vein occlusions (RVO), are leading causes of vision loss and blindness among working age adults in the United States [1]. DR and DME, AMD, and RVO together affect over 45 million people in the US, with varying prevalence rates across different demographic and socioeconomic groups [1,2,3,4,5,6,7].

Historically marginalized populations—broadly defined as racial and ethnic minorities and individuals with a lower socioeconomic status (SES)—have been reported to be disproportionately affected by many common retina diseases and also experience higher rates of disparities across other aspects of care, as shown in several landmark vision-related studies [5,8,9]. For example, the Los Angeles Latino Study found that Latino participants developed vision loss and blindness at a higher rate compared to other ethnic groups and had a DR prevalence that was nearly two-fold higher than non-Latino, White participants [10,11]. On a similar note, the Salisbury Eye Evaluation reported that Black patients with DR were four times more likely to experience vision impairment and twice as likely to have DME than White individuals [9,12,13].

Unfortunately, the disparities in disease prevalence are also reflected in clinical trial participation, with minority group enrollment in ophthalmology clinical trials across the US not appearing representative of the proportion of the population affected by the diseases being investigated [14,15,16,17]. Specifically, Black and Hispanic participants have been consistently underrepresented in clinical trials that have resulted in US Food and Drug Administration (FDA) approvals, especially compared to the proportion of individuals affected by the disease under study [16]. Analyses of DR and AMD phase 3 clinical trial participants have reported that subjects who are older, have a lower income and education level, and reside in rural communities are disadvantaged and experience higher travel burdens and geographic barriers to care than other patients [14,18,19]. As such, these patients may be less motivated to pursue available treatments and consequently experience increased disease burdens.

It is important that we, as a community of retina providers and clinical trialists, acknowledge and address these gaps in equitable patient care.

A lack of representation in clinical trials decreases the generalizability of clinical trial results, particularly between different ages, genders, and racial groups [20]. Studies have reported notable differences in drug metabolism or pharmacokinetics among older and younger adults, men and women, as well as across different racial and ethnic groups [21,22,23]. In some cases, this difference has been documented to be substantial enough for drugs or therapies to be effective for one population yet potentially toxic among other populations [21]. In addition to increasing treatment generalizability, improving participation among underrepresented groups in clinical trials can prevent low accrual rates that have caused approximately half of phase I through phase IV clinical trials to be suspended or discontinued in the past decade [20].

Although disparities in the representation of historically marginalized populations are well-documented for specific retina diseases and ophthalmology clinical trials in general, less is known regarding whether or not this gap is equally prevalent in the retina field as a whole. The current study evaluated the discrepancies that exist between the demographic and socioeconomic factors of patients who declined participation in retina-focused clinical trials compared to those who enrolled, in order to inform future clinical trial recruitment and enrollment strategies.

## 2. Materials and Methods

### 2.1. Data Collection

This retrospective cohort study was conducted between 1 January 2022, and 31 December 2022, at a single tertiary retina-care practice. Patients who were referred to at least one prospective, retina-focused clinical trial at Retina Consultants of Texas, a large, urban, retina-based practice, were identified using electronic health records. Self-reported demographic information (age, gender, race, ethnicity, and preferred language) and socioeconomic information (insurance status, social security number (SSN) status, and street address and zip code) were manually extracted using RealTime-CTMS (RealTime Software Solutions, LLC, San Antonio, TX, USA) and Nextech IntelleChartPRO (Nextech Systems, LLC, Tampa, FL, USA).

Patient recruitment status after the initial referral was categorized as Enrolled, Declined, Communication, and Did Not Qualify (DNQ). Enrolled patients were those participating in a clinical trial in 2022. Declined patients were those who were referred to a clinical trial in 2022 but elected not to participate. Communication patients were defined as those who were not contacted, were contacted with no response, were waiting for a follow-up, or were scheduled for screening following a clinical trial referral. DNQ patients were those who were referred and interested in participating in 2022, but did not meet the study criteria for the clinical trial(s) to which they were referred.

The patient’s street address and zip code were used to estimate their median household income. Median household income was identified and estimated using Geocodio (Dotsquare LLC, Norfolk, VA, USA), a batch geocoding application programming interface [24,25,26]. Geocodio extracted data from the 2021 US Census Bureau’s American Community Survey for each patient address’ Census Block Group. Patients without residential street addresses were excluded from the median household income analyses.

### 2.2. Statistical Analysis

Univariable analyses were employed to determine significant relationships between the Enrolled and Declined patients. Multivariable analyses were used to assess the strength of the association between each demographic and socioeconomic factor and patient recruitment status. Patients with no information provided were excluded from the analysis for each variable except for Insurance Status and SSN Status. Analyses were conducted using Stata (StataCorp LLC, College Station, TX, USA) and Microsoft Excel (Microsoft, Redmond, WA, USA).

## 3. Results

A total of 1477 patients were identified for inclusion between 1 January 2022, to 31 December 2022. The distribution of demographic and socioeconomic factors is shown in Table 1 and Table 2, respectively. The mean (standard deviation (SD)) age was 68.5 (13.9) years; 647 patients (43.9%) were male, 900 (61.7%) were White, 139 (9.5%) were Black, 54 (3.8%) were Asian, 365 (25.0%) were of another race, and 275 (18.7%) were Hispanic; 1306 patients (90.6%) listed English as their preferred language, 1358 (91.9%) had insurance on file, and 1273 (86.2%) reported having an SSN; the average median household income (SD) was $95,300 ($48,714). Of the 1477 patients identified, 635 (43.0%) patients formed the Enrolled group, 232 (15.7%) patients formed the Declined group, 290 (19.6%) patients formed the Communication group, and 320 (21.7%) patients formed the DNQ group.

### 3.1. Factors Statistically Significantly Associated with Clinical Trial Recruitment Status

#### 3.1.1. Age (*n* = 1476)

The mean (SD) age was 67.9 (14.2) years in the Enrolled group, 70.3 (13.8) years in the Declined Group, 66.5 (14.3) years in the Communication group, and 70.0 (12.6) years in the DNQ group. Patients in the Enrolled group were significantly older than patients in the Declined group (*p* < 0.05).

#### 3.1.2. Ethnicity (*n* = 1472)

The ethnicity distribution of the patients was 102 (16.1%) Hispanic and 466 (73.4%) not Hispanic in the Enrolled group, 42 (18.4%) Hispanic and 177 (77.6%) not Hispanic in the Declined group, 67 (23.2%) Hispanic and 196 (67.8%) not Hispanic in the Communication group, and 64 (20.0%) Hispanic and 244 (76.3%) not Hispanic in the DNQ group. Ethnicity was recorded as “Other” for 67 (10.5%), 9 (4.0%), 26 (9.0%), and 12 (3.7%) patients in the Enrolled, Declined, Communication, and DNQ groups, respectively. Comapred to the Enrolled group, the Declined group was associated with a significantly higher proportion of patients who were Hispanic (*p* = 0.01), as shown in Figure 1.

#### 3.1.3. Preferred Language (*n* = 1442)

The preferred language distribution of patients by the groups was as follows. Among the Enrolled patients, 573 (92.7%) preferred English, 39 (6.3%) preferred Spanish, and 6 (1.0%) preferred another language. Among the Declined patients, 202 (89.4%) preferred English, 23 (10.2%) preferred Spanish, and 1 (0.4%) preferred another language. Among the Communication patients, 248 (87.9%) preferred English, 33 (11.7%) preferred Spanish, and 1 (0.4%) preferred another language. Among the DNQ patients, 283 (90.4%) preferred English, 29 (9.3%) preferred Spanish, and 1 (0.3%) preferred another language. Patients in the Declined group were significantly more likely to prefer Spanish over English compared to patients in the Enrolled group (*p* < 0.05), as shown in Figure 2.

#### 3.1.4. Insurance Status (*n* = 1477)

The insurance distribution of patients by recruitment status was as follows. Among the Enrolled patients, 600 (94.5%) had insurance, and 35 (5.5%) did not have insurance on file. Among the Declined patients, 204 (87.9%) had insurance, and 28 (12.1%) did not have insurance on file. Among the Communication patients, 261 (90.0%) had insurance, and 28 (10.0%) did not have insurance. Among the DNQ patients, 293 (91.6%) had insurance, and 27 (8.4%) did not have insurance on file. Patients in the Declined group were significantly more likely to report not having insurance compared to patients in the Enrolled group (*p* = 0.001), as shown in Figure 3.

#### 3.1.5. SSN Status (*n* = 1477)

The SSN status distribution of patients was 593 (93.4%) with an SSN and 42 (6.6%) without an SSN on file in the Enrolled group, 179 (77.2%) with an SSN and 53 (22.8%) without an SSN on file in the Declined group, 233 (80.3%) with an SSN and 57 (19.7%) without an SSN on file in the Communication group, and 268 (83.8%) with an SSN and 52 (16.3%) without an SSN on file in the DNQ group. Patients in the Declined group were significantly more likely to report not having an SSN compared to patients in the Enrolled group (*p* < 0.001), as shown in Figure 4.

### 3.2. Factors Not Significantly Associated with Clinical Trial Recruitment Status

The gender distribution of patients was 300 (47.3%) male and 334 (52.7%) female in the Enrolled group, 101 (43.5%) male and 131 (56.5%) female in the Declined group, 113 (39.1%) male and 176 (60.9%) female in the Communication group, and 133 (41.6%) male and 187 (58.4%) female in the DNQ group. There was no significant difference between the gender distribution of the Enrolled and the Declined patients (*p* > 0.05).

The race distribution of patients by the groups was as follows. The Enrolled patients were 368 (59.1%) White, 67 (10.8%) Black, 27 (4.3%) Asian, and 161 (25.8%) Other. The Declined patients were 147 (64.2%) White, 18 (7.9%) Black, 13 (5.7%) Asian, 51 (22.2%) Other. Communication patients were 165 (57.5%) White, 36 (12.5%) Black, 11 (3.9%) Asian, and 75 (26.1%) Other. DNQ patients were 220 (68.8%) White, 18 (5.6%) Black, 4 (1.2%) Asian, and 78 (24.4%) Other. There was no significant difference between the race distribution of Enrolled and Declined patients (*p* > 0.05).

### 3.3. Multivariable Analysis for Demographic Factors: Odds Ratios

Male, White, Hispanic, and preferred English were used as the reference groups for the odds interpretation of gender, race, ethnicity, and preferred language, respectively. The multivariable regression model is shown in Table 3.

There were significant odds ratios between the Enrolled and Declined groups for age (*p* < 0.02, odds ratio (OR) = 0.98, 95% confidence interval (CI) [0.97, 1.00]), and between patients who preferred English versus Spanish (*p* = 0.004, OR = 0.35, 95% CI [0.17, 0.72].

The odds ratios that were not significant for the Enrolled and Declined groups included gender (*p* > 0.05, OR = 0.86, 95% CI [0.62, 1.18], race, specifically between White and Black patients (*p* > 0.05, OR = 1.23, 95% CI [0.69, 2.20], and ethnicity, specifically between Hispanic and not Hispanic patients (*p* > 0.05, OR = 0.80, 95% CI [0.44, 1.44].

## 4. Discussion

The current study evaluated demographic and socioeconomic factors that potentially had a consequential influence on patient participation in retina-focused clinical trials. Age, ethnicity, and preferred language were all found to be significant demographic factors, and insurance and SSN status were found to be significant socioeconomic factors. These results corroborate findings from other clinical trial enrollment studies [13,16,27,28,29,30,31,32,33,34,35,36,37].

The interpretation of the odds ratio for age was that a one-year increase in age was associated with 0.98 times the odds of being in the Enrolled group when all other factors were the same. The mean age of the Enrolled patients was found to be significantly lower than that of the Declined patients (*p* < 0.05). While findings from ophthalmology-specific clinical trials were relatively limited, clinical trial data from other fields similarly found that older adults were consistently underrepresented [27,28,29,30]. Cancer clinical trial data have shown that, despite two-thirds of cancer patients being over 65 years of age, only a quarter of trial participants are of this age cohort, and the gap between the median age of the clinical trial and general populations has widened over time [27,28]. Other reviews of randomized clinical trials have reported that adults over the age of 65 were disproportionately excluded, particularly from phase 3 and phase 4 trials [29,30]. The discrepancy in the age of patients who enrolled in retina-focused clinical trials could be at least partially related to the age requirements in clinical trial protocols, as age has been used as an exclusion criterion in some studies [30]. However, given that aging increases the risk of many common retina diseases, recruiting patients for retina clinical trials that accurately represent the affected population is important for the generalizability of the resultant data.

A greater proportion of patients who declined clinical trial participation identified as Hispanic compared to those who were enrolled. While the enrollment of Hispanic patients appears to have improved in both clinical trials overall and in ophthalmology-related clinical trials, disparities in enrollment are still prevalent [16,31]. The enrollment incidence disparity (EID), defined as the difference between the number of trial participants and the participants diagnosed with a condition of a particular race or ethnicity, is expected to worsen for Hispanic patients with AMD [16].

One potential reason for this significant difference in ethnicity between the Enrolled and Declined groups is the corresponding significant difference in the preferred language, with a significantly greater number of Enrolled patients preferring English; Spanish was the second most commonly preferred language. Significant relationships were observed when comparing ethnicity with preferred language distributions within both the Enrolled and Declined groups (*p* < 0.001 for Enrolled, *p* < 0.001 for Declined). Furthermore, patients who preferred Spanish had 0.44 times the odds of enrolling in a clinical trial compared to those who preferred English (OR = 0.44, 95% CI [0.21, 0.93]). It is also worth mentioning that a higher proportion of Communication patients identified as Hispanic and preferred Spanish compared to all other recruitment statuses. Language barriers could correspond with a poorer understanding of diagnoses and prognoses, leading to communication difficulties between the patient and provider and increased time spent in the clinic for clinical trial consenting [13,32,33]. Greater difficulties in accessing and receiving care may discourage patients who prefer speaking a language other than English from engaging in the clinical trial process. These findings also highlight the potential value of having clinical staff available to communicate with patients in their preferred language to enhance understanding.

Patients who enrolled in a clinical trial were more likely to have medical insurance and an SSN than those who declined. While studies on insurance and SSN status in relation to clinical trial participation are scarce, published studies have reported that patients with fewer financial resources, such as medical insurance, are often underrepresented in cancer clinical trials [34]. Uninsured individuals have also been reported to be more likely to struggle with accessing necessary medical care while simultaneously experiencing higher rates of disease onset and severity [35,36,37].

No significant differences were observed for the other variables studied in the current analysis, including gender, race, and median household income. In comparison, other studies have reported income levels to be inversely associated with trial participation likelihood [38]. Alternative methods for estimating median household income are needed to confirm whether or not the trend observed in the current study accurately reflects the patient population of interest.

Perhaps the most unexpected finding in this study was the lack of a significant difference between the race distributions of the Enrolled and Declined patients. Multiple prior studies have reported that Black patients are meaningfully underrepresented in clinical trials, even when other minority races are adequately represented [16]. Black patients have experienced a threefold disparity in representation in DME trials funded by the National Institute of Health and more than a fourfold disparity in representation in DME trials funded by the pharmaceutical industry or other federal US organizations [39]. Historically, this discrepancy can be explained in part by patient hesitancy to participate in clinical trials due to a lower level of trust in medical institutions and poorer access to healthcare [40,41]. Within the current dataset, it is noteworthy that nearly 15% of all patients referred to a clinical trial were racial minorities. A possible explanation for the lack of significant difference found in the current analysis is that patients in the current analysis were typically referred for the consideration of clinical trial participation after the physician engaged with the patient and recommended clinical trial participation directly to them.

### Limitations

There are several limitations that may impact the findings of this study, including, most notably, the availability of patient information. Insurance and SSN information was not required for patients who ultimately did not enroll in a clinical trial, so the proportion of patients who have insurance and an SSN on file may be underreported for patients whose recruitment statuses were Declined, DNQ, or Communication.

Another limitation is the accuracy of acquiring median household income using street address and zip code. Zip codes have previously served as a simple method of obtaining socioeconomic status indicators [42]. However, accuracy can be lost when using zip code and other location-based data for patients from smaller populations since US Census data are often not recorded on such a small scale [43]. Due to a lack of accuracy in prevailing methods, further investigation is necessary to gain a better understanding of the influence that median household income may have on clinical trial participation.

Furthermore, approximately one-fourth of patients in each recruitment status category did not have a reported race on file. As a result, it was difficult to determine whether or not this race distribution was an accurate reflection of the patient population.

## 5. Conclusions

There are numerous reasons why patients may not enroll in clinical trials despite physician referral, including not meeting the trial criteria, communication issues following referral, and electing to not participate. The current study focused on comparing demographic and socioeconomic factors between patients who successfully enrolled (Enrolled) with those who declined clinical trial participation (Declined). Ultimately, statistically significant differences were observed in age, ethnicity, preferred language, insurance status, and SSN status between these patient populations. An awareness of these disparities may be valuable for developing strategies that can improve access to clinical trials for all patients.

## Figures and Tables

**Figure 1 jpm-13-00880-f001:**
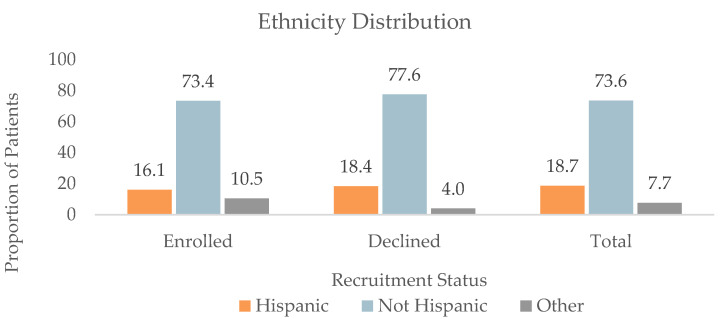
Ethnicity distribution of all patients (*n* = 1472), patients who enrolled in (*n* = 635), and patients who declined enrollment (*n* = 228) for at least one retina-focused prospective clinical trial in 2022. There were significant differences in ethnicity between the Enrolled and Declined groups (*p* < 0.0002).

**Figure 2 jpm-13-00880-f002:**
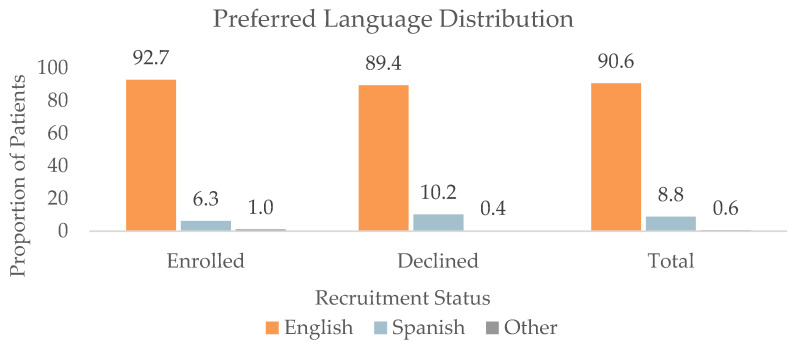
Preferred language distribution of all patients (*n* = 1442), patients who enrolled in (*n* = 618), and patients who declined enrollment (*n* = 226) for at least one retina-focused prospective clinical trial in 2022. There were significant differences in the preferred language between the Enrolled and Declined groups (*p* < 0.05).

**Figure 3 jpm-13-00880-f003:**
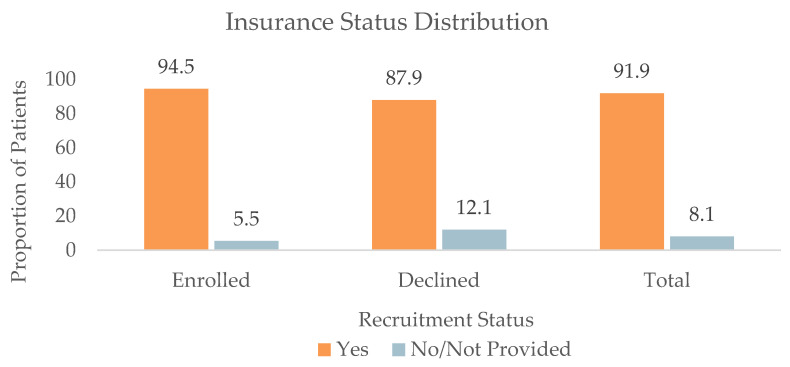
Insurance status distribution of all patients (*n* = 1477), patients who enrolled in (*n* = 635), and patients who declined enrollment (*n* = 232) for at least one retina-focused prospective clinical trial in 2022. There were significant differences in the insurance status between the Enrolled and Declined groups (*p* = 0.001).

**Figure 4 jpm-13-00880-f004:**
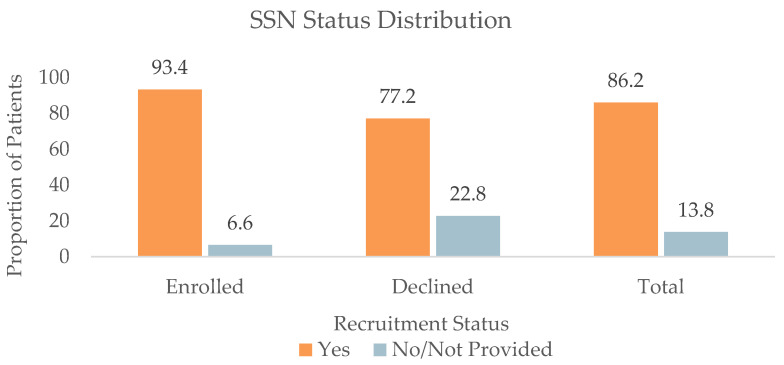
Social security number (SSN) status distribution of all patients (*n* = 1477), patients who enrolled in (*n* = 635), and patients who declined enrollment (*n* = 232) for at least one retina-focused prospective clinical trial in 2022. There were significant differences in the SSN status between the Enrolled and Declined groups (*p* < 0.001).

**Table 1 jpm-13-00880-t001:** Demographic characteristic distribution of patients referred to at least one retina-focused prospective clinical trial in 2022.

Characteristic		*n* ^1^	All Patients	Enrolled	Declined	Communication	DNQ
Total		1477	1477	635	232	290	320
Average Age (SD)		1476	68.5 (13.9)	67.9 (14.2)	70.3 (13.8)	66.5 (14.3)	70.0 (12.6)
Gender (%)	Male	1475	647 (43.9)	300 (47.3)	101 (43.5)	113 (39.1)	133 (41.6)
	Female		828 (56.1)	334 (52.7)	131 (56.5)	176 (60.9)	187 (58.4)
Race (%)	White	1459	900 (61.7)	368 (59.1)	147 (64.2)	165 (57.5)	220 (68.8)
	Black		139 (9.5)	67 (10.8)	18 (7.9)	36 (12.5)	18 (5.6)
	Asian		54 (3.8)	27 (4.3)	13 (5.7)	11 (3.9)	4 (1.2)
	Other		365 (25.0)	161 (25.8)	51 (22.2)	75 (26.1)	78 (24.4)
Ethnicity (%)	Hispanic	1472	275 (18.7)	102 (16.1)	42 (18.4)	67 (23.2)	64 (20.0)
	Not Hispanic		1083 (73.6)	466 (73.4)	177 (77.6)	196 (67.8)	244 (76.3)
	Other		114 (7.7)	67 (10.5)	9 (4.0)	26 (9.0)	12 (3.7)
Preferred Language (%)	English	1442	1306 (90.6)	573 (92.7)	202 (89.4)	248 (87.9)	283 (90.4)
	Spanish		126 (8.8)	39 (6.3)	23 (10.2)	33 (11.7)	29 (9.3)
	Other		9 (0.6)	6 (1.0)	1 (0.4)	1 (0.4)	1 (0.3)

^1^ Varying sample sizes (*n*) reflect the number of patients with information available for the specified characteristic.

**Table 2 jpm-13-00880-t002:** Socioeconomic characteristic distribution of patients referred to at least one retina-focused prospective clinical trial in 2022.

Characteristic		All Patients	Enrolled	Declined	Communication	DNQ
Total		1477	635	232	290	320
Insurance Status (%)	Yes	1358 (91.9)	600 (94.5)	261 (90.0)	204 (87.9)	293 (91.6)
	No/not provided	119 (8.1)	35 (5.5)	28 (10.0)	28 (12.1)	27 (8.4)
SSN (%)	Yes	1273 (86.2)	593 (93.4)	233 (80.3)	179 (77.2)	268 (83.8)
	No/not provided	204 (13.8)	42 (6.6)	57 (19.7)	53 (22.8)	52 (16.3)

**Table 3 jpm-13-00880-t003:** Multivariable regression (odds ratios) of demographic factors between Enrolled and Declined groups. Male, White, Hispanic, and preferred English were used as the reference groups for the odds interpretation of gender, race, ethnicity, and preferred language, respectively.

Characteristic		Odds Ratio	Standard Error	z	*p*	[95% Confidence Interval]	
Age		0.98	0.007	−2.47	0.01	0.97	1.00
Gender	Female	0.86	0.14	−0.95	0.34	0.62	1.18
Race	Black	1.23	0.36	0.69	0.49	0.69	2.20
	Asian	1.14	0.27	0.57	0.57	0.72	1.81
	Other	0.62	0.23	−1.27	0.20	0.30	1.29
Ethnicity	Not Hispanic	0.80	0.24	−0.75	0.45	0.44	1.44
	Other	2.34	1.19	1.66	0.10	0.86	6.36
Preferred language	Spanish	0.35	0.13	−2.86	0.00	0.17	0.72
	Other	3.1	3.43	1.01	0.32	0.34	27.35
Baseline Odds		11.18	5.61	4.81	0.00	4.18	29.89

## Data Availability

The data that support the findings of this study are available upon request from the corresponding author. The data are not publicly available due to privacy or ethical restrictions.

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
