# Peer review of "Demographic and Socioeconomic Factors in Prospective Retina-Focused Clinical Trial Screening and Enrollment"

_jpm, 2023, doi:10.3390/jpm13060880_

Round 1
Reviewer 1 Report
This is a very well conducted study and well written paper which has helped to highlight the disparities that can occur in clinical trial participation due to various demographic and socioeconomic factors. It is probably beyond the scope of this paper, but a further study looking more in depth at other factors that might contribute this e.g. unconscious biases in recruiting personnel, the accessibility of trial recruitment materials that were used, availability of funding to compensate for participants time etc and exploring factors that could help mitigate this problem through a systematic literature review and the conduct of focus group discussions etc would be very interesting.
Reviewer 2 Report
The authors sought to explore a patient population that aimed to recruit them for a retina study and determine whether the recruitment methods resulted in in various patient parameters such as age, ethnicity, etc. There are no comments from this peer reviewer. The statistical methodology is good and sound. The data presentation is good. The findings are important to advance the recruitment of patients for retina-based studies and for future studies to keep in mind when recruiting patients.